# Finding Dr. Kim: Information Sources of Korean Immigrants’ Search for a Doctor in the U.S.

**DOI:** 10.3390/healthcare8020092

**Published:** 2020-04-09

**Authors:** Sou Hyun Jang, Linda K. Ko, Hendrika Meischke

**Affiliations:** 1Department of Sociology, Sungkyunkwan University, Seoul 03063, Korea; 2Department of Health Services, University of Washington, Seattle, DC 98195, USA; lko@fredhutch.org (L.K.K.); hendrika@uw.edu (H.M.); 3The Fred Hutch Cancer Research Center, Public Health Sciences Division, Seattle, DC 98109, USA

**Keywords:** searching for a doctor, health information seeking, Korean immigrants, co-ethnic doctors, recent immigrants, information sources

## Abstract

Korean immigrants in the United States (U.S.) are known for their preference for, and dependence on, co-ethnic doctors due to various barriers to the U.S. healthcare system. Recent immigrants tend to face more barriers than their non-recent counterparts. However, there is little information on how they find their doctors in the U.S. This study includes a self-administrated survey of Korean immigrants aged 18 and above who lived in the New York–New Jersey Metropolitan area in 2013–2014 (*n* = 440). Descriptive analysis was conducted to understand the most common information sources and the number of sources based on the duration of stay in the U.S. More recent Korean immigrants were female, had no family doctor, uninsured, younger, and more educated than their non-recent counterparts. Regardless of the duration of stay in the U.S., family members and friends were the most frequently sought-after sources for Korean immigrants in their search for doctors. In addition to family members and friends, non-recent Korean immigrants also used other methods (e.g., Korean business directories), whereas recent immigrants used both U.S. and Korean websites. More recent Korean immigrants used multiple sources compared to non-recent Korean immigrants, often combined with a Korean website. Our study suggests policy implications to improve recent immigrants’ accessibility to health information in a timely manner.

## 1. Introduction

When immigrants leave their home country, they also leave their doctors; this implies the search for a new doctor upon arriving in the destination country. Not only recent immigrants, but also non-recent immigrants may be obliged to find new doctors if they are not satisfied with the treatment offered by their current doctor, or if their current doctor is no longer available. Both recent and non-recent immigrants tend to prefer co-ethnic doctors, who share the same language and culture of their home country, as they often face challenges navigating the U.S. healthcare system, due to language barriers and cultural differences [1,2]. Previous studies have also found that recent immigrants are likely to face more barriers in accessing American healthcare—thus underutilizing it—compared with their non-recent counterparts [3,4,5,6]. 

Korean immigrants are one of the most rapidly increasing ethnic minority groups and the fifth- largest Asian group in the United States (U.S.) [7,8]. They also prefer, and are likely to see, co-ethnic doctors, mainly due to their uninsured status, language barriers, cultural differences, and limited knowledge of Western medicine [1,9,10]. Additionally, recent Korean immigrants have a higher likelihood of seeking healthcare from their ethnic community to help navigate the U.S. healthcare system, which is based on a multi-payer and a referral system, as opposed to the National Health Insurance Program in Korea [9]. 

Health information-seeking behaviors are important, as it is closely related to individuals’ health-related decision-making, including healthcare access [11,12]. Researchers who have examined various types of health information-seeking behaviors among Korean immigrants, including those related to prescription drugs [13], diabetes [14], and cancer [15], have highlighted that Korean immigrants utilize various sources to seek health information, such as healthcare professionals, printed materials, and mass media sources [13,14]. An earlier study found that Korean immigrants’ health information-seeking behaviors are heavily dependent on ethnic media, including Korean newspapers and magazines [13]. More recently, in the era of the internet, ethnic online communities have been playing an important role in helping Korean immigrants who are seeking health information [14,15].

However, despite a plethora of studies that examined Korean immigrants’ preference for co-ethnic doctors and health information-seeking behaviors, no study has so far examined how Korean immigrants find their doctor: Dr. Kim. This study uses the fictitious name Dr. Kim, since Kim is the most popular and widely recognized last name among Koreans, accounting for approximately 20% of all Korean last names [16]. This study has two purposes. First, it examines the information sources that Korean immigrants use to search for their doctors in the U.S. Second, it investigates whether these information sources differ in accordance with Korean immigrants’ duration of stay in the U.S. 

## 2. Materials and Methods

### 2.1. Data and Participants 

This cross-sectional study used a self-administrated survey questionnaire. We collected data from Korean immigrants aged 18 and above in the New York/New Jersey Metropolitan area. Korean immigrants who have been living in the U.S. for less than a year were excluded from the survey. We used a convenience sampling approach to recruit participants through ethnic churches, community centers, and ethnic festivals between September 2013 and June 2014. We obtained information from 507 Korean immigrants; however, only 440 Korean immigrants, 86.8% of the entire sample, have talked to, or seen, a doctor in the last five years. Therefore, the final sample included only 440 participants. There were 40 questions, asking participants about their socio-demographic characteristics, current health conditions, health insurance status, preferred characteristics of family doctors, and how they find a doctor in the U.S. This study was approved by the Institutional Review Board at the City University of New York (491073-2).

### 2.2. Measures 

Participants were asked to identify the sources that they used to find a doctor in the U.S. Responses included: (1) family members, (2) friends, (3) co-workers, (4) church members, (5) U.S. websites (e.g., Google, Yelp, and ZocDoc), (6) Korean websites (e.g., Naver, Heykorean, and MissyUSA), and (7) other methods (e.g., Korean business directories, ethnic newspaper advertisements, and signboards on streets). The participants were also asked to indicate the duration of their stay in the U.S.: (1) 1–2 years, (2) 3–4 years, (3) 5–9 years, (4) 10–19 years, and (5) 20 years or more. To compare information sources by duration of stay in the U.S., we split Korean immigrants into two groups: (1) recent immigrants (lived in the U.S. for less than 10 years), and (2) non-recent immigrants (lived in the U.S. for 10 years or more). Previous studies have also used the 10-year mark, and we found this to be significant in relation to healthcare utilization in the U.S. [17,18]. We also included immigration status (e.g., temporary visa, working visa, permanent resident, U.S. citizenship, and others), although more than half of the participants (228/440 = 51.8%) refused to reveal it.

Adapting questions from the New Immigrant Survey [19], we measured age categories (18–29, 30–39, 40–49, 50–64, and 65 and above), gender (male vs. female), and level of education (high school or below, some college or college graduates, and higher than Bachelor’s degree). We measured English proficiency by asking participants how well they speak English: (1) not at all, (2) a little, (3) well, and (4) very well. We combined responses into not at all/a little and well/very well for analysis. For insurance status, participants chose from the following options: (1) private health insurance, (2) employer/union health insurance, (3) Medicaid, (4) Medicare, and (5) other. Participants qualified their health status on the following scale: (1) bad, (2) not bad, (3) good, and (4) very good. We combined bad/not bad and good/very good for analysis. Finally, we asked participants about their family doctor in the U.S.: (1) none, (2) non-Korean, (3) first-generation Korean, and (4) 1.5- or second-generation Korean.

### 2.3. Data Analysis

We used a Chi-squared test to identify the difference in characteristics between recent and non-recent Korean immigrants, and conducted descriptive analysis to understand the most common information sources and the number of sources to search for doctors by duration of stay in the U.S. All analyses were done in Stata 15.0. (StataCorp LLC, College Station, TX, USA)

## 3. Results

### 3.1. Characteristics of Participants 

Table 1 shows the characteristics of the participants. About one-third were aged 18–39 years (33.4%), about a quarter were 40–49 years (24.1%) and 50–64 years (23.6%), and about one-fifth were 65 years or above (18.9%). Females accounted for more than half of the sample (56.4%). Slightly more than a quarter (26.3%) had a high school education or lower, while the rest had a college education or higher. More than half (54.1%) reported that they do not speak English well, and many were insured (72.9%) and reported excellent health status (68.9%). Slightly less than one-third (30.7%) had no family doctor in the U.S., and among those who had one, the majority (85.9%) had a co-ethnic doctor. 

There were statistically significant differences in age, gender, level of education, immigration status, health insurance status, and having family doctors, based on the participants’ duration of stay in the U.S. Recent immigrants were more likely to be younger, female, and with slightly higher levels of education than non-recent immigrants. Among recent immigrants, 20.9% had a temporary visa, 5.5% had a working visa, 17.6% permanent residency, 3.3% U.S. citizenship, and 49.5% with no response, while the figures among non-recent immigrants were 0.3%, 1.4%, 15.5%, 32.7%, and 47.8%, respectively. In addition, recent immigrants were less likely to be insured than their non-recent counterparts (60.4% vs. 76.2%, respectively), and were considerably more likely to not have a family doctor than non-recent immigrants (61.5% vs. 22.6%). 

### 3.2. Types and Number of Sources of Korean Immigrants’ Search for Doctors 

Table 2 shows the types and numbers of information sources that Korean immigrants use to search for doctors in the U.S., based on the duration of their stay in the U.S. Among the participants, family members (35.2%) were the most popular source, followed by friends (27.3%), co-workers (17.7%), other methods (16.4%), U.S. websites (13.6%), church members (10.0%), and Korean websites (8.9%). While most (80.5%) used a single source to search for doctors, about one-fifth (19.6%) utilized multiple sources. Among those who used multiple sources, two most frequently used methods were a combination of two interpersonal sources, such as family members and friends (10.5%) and family members and co-workers (8.1%).

While family members and friends remained the most popular sources for both recent and non-recent Korean immigrants, the use of other sources differed between the groups. Co-workers (17.8%), other methods (17.2%), U.S. websites (10.6%), and church members (9.5%) were popular sources for non-recent Korean immigrants’ search for doctors, while Korean websites were the least popular (< 7%). Websites—both U.S. (25.3%) and Korean (16.5%)—were major sources in recent Korean immigrants’ search for doctors in the U.S. A lower proportion of recent Korean immigrants used other methods (13.2%) compared with their non-recent Korean immigrant counterparts (17.2%). 

Both groups also differed on the number of information sources utilized. Less than one-fifth of non-recent Korean immigrants (17.5%) utilized multiple sources while looking for a doctor, compared with more than a quarter of recent Korean immigrants (27.5%). Although Korean websites were not the most popular sources among recent immigrants, participants liked to combine them with interpersonal sources. When non-recent Korean immigrants utilized multiple sources, family members were most commonly used, in combination with other sources. 

## 4. Discussion

The current study is one of the first studies to investigate Korean immigrants’ health information-seeking behavior by descriptively highlighting the information sources Korean immigrants use to search for doctors in the U.S. We found that family members and friends were the most frequently used sources, regardless of the duration of stay in the U.S. This finding is in line with an earlier study that also found family members and friends to be the most used sources for seeking health information among Korean immigrants, primarily due to language support, cultural cohesion, and strong trust in their closest interpersonal sources [20,21]. In addition to family members and friends, we found that Korean immigrants also used co-workers, U.S. and Korean websites, church members, and other sources. 

The sources of searching for a doctor differed based on Korean immigrants’ duration of stay in the U.S.; while recent Korean immigrants were more likely to utilize internet websites, non-recent Korean immigrants used interpersonal and other sources (e.g., Korean media business directories). We also found that more recent Korean immigrants, compared to non-recent Korean immigrants, utilized multiple sources, particularly relying on Korean websites to find their doctors in the U.S. One possible explanation for this difference between recent and non-recent groups is their social network in the U.S. Recent immigrants have fewer interpersonal social networks in the new country [20,21]; thus, they may be less networked with reliable interpersonal sources to find doctors. Instead, they might be engaged in seeking health information online as a fallback; furthermore, they may utilize multiple sources to confirm the credibility of the information searched online. Another possible explanation may be related to age and education. Recent immigrants are relatively younger and with slightly higher levels of education than their non-recent counterparts; thus, they may be comfortable and familiar with internet-based information, corroborating results from other studies [22].

This descriptive study has certain limitations, including generalizability. Specifically, our study used a convenience sampling approach, limiting our ability to generalize the results across Korean immigrants in the New York/New Jersey Metropolitan area. However, convenience sampling is a common approach used in studies on immigrant communities [14,23]. Additionally, the survey questionnaire did not include questions on whether participants had face-to-face interaction or utilized information and communication technologies (ICT) to obtain information about a doctor via their family, friends, co-workers, and church members, since our goal was to understand information sources in general and not how Korean immigrants obtained health information from their sources. Future studies could explore the role of ICT in how immigrants obtain health information. Lastly, our study was limited in scope to a cross-sectional design. A future study with a longitudinal study design will enable us to study trends of health-seeking behaviors across immigration status.

Nonetheless, it opens a door to various research opportunities for future studies. First, future studies could examine whether, and to what extent, factors such as health insurance, language barriers, size of social networks, and transnational ties with the home country are associated with Korean immigrants’ search for doctors in the U.S. Second, although earlier studies [13,14] found that Korean Americans are less likely to use mass media and printed materials to seek health information compared with white Americans, there are few studies comparing different racial/ethnic immigrant groups’ doctor-searching behaviors with that of U.S.-born individuals. Previous studies revealed that immigrants can benefit from seeing co-ethnic doctors [2], although those who visit co-ethnic doctors (including Vietnamese and Mexican immigrants), may experience additional power struggles [24]. However, to the best of our knowledge, there has been no study that examined how different immigrant groups find their co-ethnic doctor in the U.S. To continue to grow the knowledge on this topic, a future research study could identify and compare factors associated with different racial/ethnic groups’ search for doctors in the U.S. Lastly, it is imperative for future studies to examine the association between the utilization of different information sources to find doctors and health-related outcomes, such as healthcare utilization. 

Our research reaffirms the findings of previous studies, which found that recent immigrants tend to face more barriers to the U.S. healthcare system [3,4,5,6] and limited social networks [20], compared with their non-recent counterparts, suggesting that these impediments could be the reason why they actively use internet websites to find their doctors in the U.S. There is a policy implication that public health professionals assisting Korean immigrants in finding co-ethnic doctors should point Korean immigrants to ethnic online communities or educate them about other Korean immigrant-friendly sources, where they may easily find co-ethnic doctors that meet their preferences and needs. Ethnic online communities such as MissyUSA, which is the largest ethnic online community comprising about 320,000 Korean immigrant women [15,25], could be a potential platform for providing Korean immigrants access to the required information to find Dr. Kim in the U.S. 

## 5. Conclusions

In conclusion, this study contributes to the limited literature on one type of health information-seeking behavior among Korean immigrants, by focusing on how they find doctors in the U.S. We believe that our findings set forth a policy implication to help recent immigrants access healthcare information more easily and in a timely manner. We hope that our findings will serve as a preliminary study for future academic studies, as well as for the Korean community in the U.S. 

## Figures and Tables

**Table 1 healthcare-08-00092-t001:** Characteristics of Korean immigrant survey participants.

Characteristics	All (*n* = 440)		Non-Recent Immigrants (*n* = 349)		Recent Immigrants (*n* = 91)	
	*n*	%	*n*	%	*n*	%
**Age ***						
18–29	67	15.2	37	10.6	30	33.0
30–39	80	18.2	52	14.9	28	30.7
40–49	106	24.1	86	24.6	20	22.0
50–64	104	23.6	95	27.2	9	9.9
65 or older	83	18.9	79	22.7	4	4.4
**Gender ***						
Male	192	43.6	161	46.1	31	34.1
Female	248	56.4	188	53.9	60	65.9
**Level of Education ***						
High school or below	116	26.3	93	26.7	23	25.3
Some college/college graduates	255	58.0	212	60.7	43	47.2
Higher than Bachelor’s	69	15.7	44	12.6	25	27.5
**English Proficiency**						
Not well/a little	238	54.1	191	54.7	47	51.7
Well/very well	202	45.9	158	45.7	44	48.3
**Immigration Status ***						
Temporary visa	20	4.5	1	0.3	19	20.9
Working visa	10	2.3	5	1.4	5	5.5
Permanent resident	70	15.9	54	15.5	16	17.6
U.S. citizenship	117	26.6	114	32.7	3	3.3
Other	11	2.5	8	2.3	3	3.3
No response	212	48.2	167	47.8	45	49.5
**Health Insurance Status ***						
Uninsured	119	27.1	83	23.8	36	39.6
Insured	321	72.9	266	76.2	55	60.4
**Health Status**						
Bad/not bad	137	31.1	102	29.2	35	38.5
Good/very good	303	68.9	247	70.8	56	61.5
**Family Doctor ***						
None	135	30.7	79	22.6	56	61.5
Korean	261	59.3	230	65.9	31	34.1
Non-Korean	44	10.0	40	11.5	4	4.4

U.S.—United States; * indicates statistically significant difference between non-recent and recent Korean immigrants (*p* < 0.05).

**Table 2 healthcare-08-00092-t002:** Information sources of Korean immigrants’ search for doctors in the U.S.

Information Sources	All Immigrants (*n* = 440)	*n*	%	Non-Recent Immigrants (*n* = 349)	*n*	%	Recent Immigrants (*n* = 91)	*n*	%
Type	Family members	155	35.2	Family members	129	37.0	Family members	26	28.6
	Friends	120	27.3	Friends	95	27.2	Friends	25	27.5
	Co-workers	78	17.7	Co-workers	62	17.8	US website	23	25.3
	Other	72	16.4	Other	60	17.2	Co-workers	16	17.6
	US website	60	13.6	US website	37	10.6	Korean website	15	16.5
	Church members	44	10.0	Church members	33	9.5	Other	12	13.2
	Korean website	39	8.9	Korean website	24	6.9	Church members	11	12.1
Number	Single	354	80.5	Single	288	82.5	Single	66	72.5
	Multiple	86	19.6	Multiple	61	17.5	Multiple	25	27.5

Note: Multiple answers were possible.

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
