# Peer review of "Finding Dr. Kim: Information Sources of Korean Immigrants’ Search for a Doctor in the U.S."

_healthcare, 2020, doi:10.3390/healthcare8020092_

Round 1
Reviewer 1 Report
The title of this study has aroused my curiosity. After I got the manuscript, I wanted to quickly figure out what "Dr. Kim" was. So, I'm pretty sure this is a very fascinating subject. However, based on the rigor and integrity of an academic study, I give the following suggestions for your reference.
The authors designed a cross-sectional research design in order to understand what Korean immigrants’ sources of doctor information are when they face natural language and cultural barriers in American life.
- The list of keywords is not precise enough to reflect the value and focus of this research, and I recommend that the authors reconsider.
- In "Abstract", the author states that this study collected 440 Korean immigrants as study participants, but in "Data and participants" mentioned that "…The dataset included 507 Korean immigrants, ......... this study focus on 440 Korean immigrants who .... "", I would like to know what the turning point is. The author did not explain in this study. I suggest the author to elaborate on the population, the rules of the sample size, the sample size, and the number and percentage of valid samples.
- I keep question marks about the representativeness of the samples in this study. The evidence cited by the author is not the latest data, please update.
- Do immigrants of other ethnic group have the same problem except Korean immigrants?
- Please provide survey questions, metrics or scales for the questions, and reference sources for these questions. For example: How to investigate health status? measure? Was a sample of immigrants less than one year excluded from this study? Type of immigration? Permanent immigration? Temporary residence?
- In Table 1, the data of English proficiency is wrong, please check.
- Do the family and friends in your source of information refer to face-to-face inquiries, or do you inquire through ICT? I think the content of the question needs to be clearly defined.
- From your research results, English proficiency is not significant, but the main impact of English proficiency is confirmed by previous studies. Since the results at this stage are different from previous studies, it is necessary to propose a convincing discussion, please add further.
Reviewer 2 Report
This is a very interesting manuscript and the title is provocative.
However, there are some issues that need to be addressed.
- The introduction does not seem to define co-ethnic doctors (line 36). It would be beneficial to readers to have this definition presented.
- The last sentence in the second paragraph of the introduction requires additional information. "Additionally, recent Korean immigrants tend to seek healthcare...because of challenges posed by the new healthcare system [9]" What are these challenges? Perhaps elaborate, "such as ____"
- The abstract states "We received questionnaires from 440 Korean immigrants...". Do you mean to state "We received responses to a survey questionnaire"? If this was indeed a survey, how many items were in the survey? E.g. 20-item survey?
- I think that line 51 could be revised further. Example: "Dr. Kim is used as a pseudonym because (Kim by far is the most popular [and widely recognized] surname among Koreans). Also, is there a citation to support this claim?
- Line 52 seems awkward on the first reading. “The study has two purposes: first, it examines the sources Korean immigrants use to search for their doctors”
Perhaps it may be rephrased to state “The study has two purposes: first, it examines the resources used by Korean immigrants to search for their doctors”
- There seems to be a grammatical error in paragraph 2.1. “This is a cross-sectional study. Survey data from Korean immigrants aged 18 and above in the New York/New Jersey area were…”
There needs to be consistency is past vs. present tense.
- Line 61 mentions a survey. Is the survey appended to the manuscript? What questions were asked in the survey?
- On page 3, it states “Recent immigrants were considerably more likely to not have a family doctor than non-recent immigrants (61.5% vs. 22.5%)”. This is clear. However, data about the level of education seems less clear cut, despite this finding being statistically significant. For example, non-recent immigrants had 60.7+12.6=73.3% high level of education whereas recent immigrants had 47.2+27.5=74.7%. This difference seems too modest to claim, in line 97, that “Recent immigrants were more likely to be …highly educated than non-recent immigrants”. Please explain how you came to this conclusion. Perhaps rephrase to suggest “ with slightly higher levels of education ”?
- Table 2 has an error. The top line says 34.0% however, this should be 37.9% (129/349). Please confirm.
- Line 130 “This finding aligns to”, perhaps state “This finding aligns with”
- Line 130-131 does not make sense as to why recent immigrants relied on family and friends: “…an earlier study that also found family members and friends to be the most used sources for seeking health information …mainly due to language barriers, cultural differences, and strong trust…”
Shouldn’t family and friends be used mainly due to language support, cultural cohesion, and strong trust...”?
- Line 135 mentions sources “The sources differed”. Sources of what? Sources of health-seeking behaviors?
- Line 145 again mentions “Recent immigrants are younger and more educated”. Please see my comment 8 above if this needs to be revised.
- Line 148 mentions limitations. Are there any other methodological limitations? This study is not longitudinal…longitudinal studies might be helpful given that immigrants would be followed over a period of time…
- Line 155 seems to have a grammatical error. It should be “behaviors” not “behavior” because there are a number of doctor-searching behaviors, not a single one.
- Lines 160 to 164 is a run-on sentence, it is extremely long. Please revise.
- Line 167 ends with “could be a possible platform for this”. What does “this” refer to? Health seeking behaviors? Please revise.
- Line 169 should state “this study contributes to the limited literature on one type of health information-seeking behavior among…”. This study’s focus was about finding a family doctor, yes? There are certainly more health seeking behaviors out there e.g. seeking online tools about health-based information, googling health symptoms, etc.
- How was consent to participate in the study obtained? In writing? Line 64-65 merely mentions that the study was approved from IRB.
Please revise the manuscript with tracked changes and upload this version so that the reviewer may see the changes clearly.
Round 2
Reviewer 1 Report
Thank you for your efforts, this version has improved a lot. I only have two questions left that the author needs to strengthen.
- The questionnaire and scale presentation format need to be simplified to facilitate the reader's quick understanding.
- There are format problems in some parts of the text, for example, there are two "," at the same time. The authors needs to check the fulltext again.
Reviewer 2 Report
This is a much improved version of the manuscript.
My final recommendations are as follows:
- The abstract states "Korean immigrants in the U.S. are known for [...] due to various barriers to U.S. healthcare system."
It looks like there is a word missing after "barriers to".
Perhaps it should read " Korean immigrants in the U.S. are known for [...] due to various barriers to the U.S. healthcare system".
- Line 45 states “Health information-seeking behavior is important, as it is closely related to…”
This needs to be revised due to grammar and style. There is not a single health information-seeking behavior. Rather there are numerous ones. It might be better to perhaps state “Health information-seeking behaviors are important, as they are closely related to…”
- Line 51 states “health information-seeking behavior is heavily dependent on ethnic media…”
Please see my comment 2 above. There is not a single health information-seeking behavior. Rather there are numerous ones. This sentence may need to be revised to state “health information-seeking behaviors are heavily dependent on ethnic media…”
- The organization of Table 1 can be improved further for Immigration statusa*. You must add another line under “Other” to state “No Response” and indicate the appropriate frequency and percent. See example below:
Age (n=183) |
Frequency |
Percent |
< 30 years |
53 |
29 |
31-40 years |
51 |
27.9 |
41-50 years |
43 |
23.5 |
> 51 years |
31 |
17 |
No response |
5 |
2.7 |
Total |
183 |
100 |
Taken from: Zaheer, S. (2017). Understanding the Impact of Safety Climate, Teamwork Climate, and Mindful Organizing on Safety Outcomes at a Large Community Hospital - A Mixed-Methods Study. Unpublished doctor dissertation. Toronto: York University.
- Table 1 can be improved further, under Immigration statusa*. The frequencies for Immigration statusa* do not correspond to the denominator for that column under which they have been listed. For example, 1/349 = .286%, not .5% . 5/349 = 1.43%, not 2.8%.
As tabulated, it seems as though the denominator is (n=349). You may wish to revise according to the example I have indicated in my comment 4 above, and then in the write-up, you can indicate that you are using a different denominator (n=228). See my comment 6 below.
- For lines 125 to 127, which currently states “Among recent immigrants, 41.3% had temporary visa, 10.9% had working visa, 34.8% permanent residency, and 6.5% U.S. citizenship, while the figures among non-recent immigrants were 0.5%, 2.8%, 29.7%, and 62.6%, respectively.”
You may wish to restate as follows, “Among the participants who indicated immigration status (n=228),41.3% had temporary visa, 10.9% had working visa, 34.8% permanent residency, and 6.5% U.S. citizenship, while the figures among non-recent immigrants were 0.5%, 2.8%, 29.7%, and 62.6%, respectively.”
- Line 215, the conclusion needs to be in a separate line. Currently it is merged with the preceding paragraph and preceding section.
Please revise the manuscript with tracked changes and upload this version so that the reviewer may see the final changes clearly.
